

# Cranial bone histology of *Metoposaurus krasiejowensis* (Amphibia, Temnospondyli) from the Late Triassic of Poland

Kamil Gruntmejer[1,2], Dorota Konietzko-Meier[1,2,3] and Adam Bodzioch[1]

[1] Department of Biosystematics, University of Opole, Opole, Poland
[2] European Centre of Palaeontology, University of Opole, Opole, Poland
[3] Steinmann Institute, University of Bonn, Bonn, Germany

## ABSTRACT

In this study, 21 skull bones of *Metoposaurus krasiejowensis* from the Late Triassic of Poland were investigated histologically. Dermal bones show a diploë structure, with an ornamented external surface. The ridges consist of mostly well vascularized parallel-fibered bone; the valleys are built of an avascular layer of lamellar bone. The thick middle region consists of cancellous bone, with varying porosity. The thin and less vascularized internal cortex consists of parallel-fibered bone. The numerous Sharpey's fibers and ISF are present in all bones. The cyclicity of growth is manifested as an alternation of thick, avascular annuli and high vascularized zones as well as a sequence of resting lines. The detailed histological framework of dermal bones varies even within a single bone; this seems to be related to the local biomechanical loading of the particular part of the skull. The dynamic processes observed during the ornamentation creation indicate that the positions of the ridges and grooves change during growth and could be a specific adaptation to changing biomechanical conditions and stress distribution during bone development. In the supratemporal, the cementing lines show that the remodeling process could be involved in the creations of sculpture. The common occurrence of ISF suggests that metaplastic ossification plays an important role during cranial development. Endochondral bones preserved the numerous remains of calcified cartilage. This indicates that ossification follows a pattern known for stereospondyl intercentra, with relatively slow ossification of the trabecular part and late development of the periosteal cortex. The large accumulation of Sharpey's fibers in the occipital condyles indicates the presence of strong muscles and ligaments connecting the skull to the vertebral column.

## INTRODUCTION

Metoposaurids were large, up to 3 meter long, late Triassic Temnospondyli with a strongly dorso-ventrally flattened body, adapted to aquatic life. The most characteristic and best known part of the *Metoposaurus* skeleton is the extremely flat and parabolic skull with anteriorly located orbits (e.g., *Schoch & Milner, 2000*). The temnospondyl skull functionally represents one skeletal element; however, anatomically, it is a conglomerate of numerous

Corresponding author
Kamil Gruntmejer,
gruntmejerkamil@gmail.com

bones varying in shape and thickness, having various functions and biomechanical loads (i.e., *Fortuny et al., 2011*; *Fortuny et al., 2012*).

The flat bones of the skull represent the dermal bones that develop via direct transformation of preexisting connective tissue (*Francillon-Vieillot et al., 1990*). The external surface of the dermal bones is characteristically ornamented. A network of raised, reticulate ridges that enclose approximately flat-bottomed, interlocking, polygonal cells is the most common type. The vast majority of these cells are four-, five-, or six-sided, creating a honeycomb- or waffle-iron-like texture. In some temnospondyls, this is essentially the only texture present. The second texture type comprises raised, parallel to sub-parallel ridges separated by round-bottomed grooves (*Rinehart & Lucas, 2013*). The function of the ornamentation is still unclear. The best supported hypotheses suggest that they increase the surface area for skin supports, increase the strength of the bone, protect blood vessels or assist in thermal exchange (summarized in *Coldiron, 1974*; *Witzmann, 2009*; *Rinehart & Lucas, 2013*; *Clarac et al., 2015*; *Clarac et al., 2016*).

The histology of amniote osteoderms is well known and studied in detail for several groups (e.g., *Scheyer & Sander, 2004*; *Vickaryous & Sire, 2009*; *De Buffrénil et al., 2011*; *Burns, Vickaryous & Currie, 2013*; *Scheyer, Desojo & Cerda, 2014*; *Cerda et al., 2015* and further references in all). The histology of temnospondyl dermal bones is less known and was first described by *Gross (1934)*, who provided a short description of the skull bones of *Mastodonsaurus*, *Metoposaurus* and *Plagiosternum*, and recognized that the dermal bones exhibit a diploë structure. Later histological studies on the dermal bones in Temnospondyli have focused mainly on morphology and vascular network and collagen fiber organization (*Bystrow, 1947*; *Enlow & Brown, 1956*; *Coldiron, 1974*; *De Ricqlès, 1981*; *Castanet, Francillon-Vieillot & De Ricqlès, 2003*; *Scheyer, 2007*), and were limited only to a few taxa. The systematic studies of dermal bones within numerous tetrapod taxa were provided by *Witzmann (2009)* and *De Buffrénil et al. (2016)*. Up until now, the dermal bones of *Metoposaurus* have not been studied in detail histologically. The only record of the histological description of *Metoposaurus diagnosticus* dermal bone was given by *Gross (1934)*, later re-described by *Witzmann (2009)*. However, it is unclear if the illustrated section was derived from the skull or the pectoral girdle, or even if the published bone fragment belongs to *Metoposaurus* at all.

The main goal of this study is to present a detailed description of the histology of dermal and endochondral bones from one skull of *Metoposaurus krasiejowensis* (*Sulej, 2002*), and determine, if possible, the tendencies and variability of the histological framework. Moreover, the value of the dermal bones for the skeletonchronological analyses, the ossification modes of the skull, and origin of the sculpture will be evaluated.

## MATERIAL AND METHODS

### Material

The skull (UOPB 01029; 40 cm in length) of *Metoposaurus krasiejowensis* was studied histologically (Figs. 1 and 2). The roof side of the skull was almost completely preserved, whereas on the palatal side only the fragments of the vomer, parasphenoid, pterygoids, quadrates and exoccipitals were preserved. The species discovered in Poland was originally

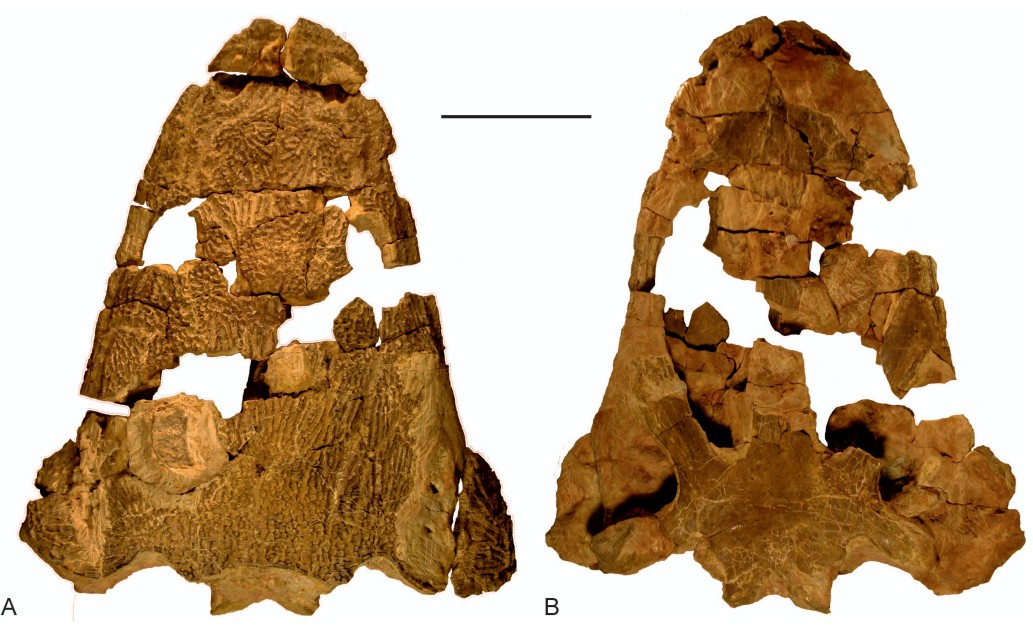

**Figure 1** **The skull of *Metoposaurus krasiejowensis* (UOPB 01029) from the Late Triassic of Poland.** (A) Dorsal view of skull; (B) Ventral view of the skull. Scale bar equals 10 cm.

described as *Metoposaurus diagnosticus krasiejowensis Sulej, 2002*, the subspecies of an older *Metoposaurus diagnosticus* (*Von Meyer, 1842*). *Brusatte et al. (2015)* recommended against using the subspecies for the German *M. diagnosticus diagnosticus* (*Von Meyer, 1842*) and the Polish *M. diagnosticus krasiejowensis*, suggesting instead to separate them at the species level as *M. diagnosticus* and *M. krasiejowensis*. That taxonomy is followed in this study.

## Locality

The examined material comes from the famous locality in Krasiejów where a large number of disarticulated skeletons were discovered in the Upper Triassic (Keuper), fine-grained, continental sediments. The bones can be found in two main bone-bearing horizons, referred to as the lower and the upper horizon (*Sulej, 2007*; *Dzik & Sulej, 2007*). The lower horizon was deposited on an alluvial plain during a catastrophic mud-flow event (*Bodzioch & Kowal-Linka, 2012*). The skull presented here was excavated from the less than 1 m thick lower bone-bearing layer, which is very rich in *Metoposaurus krasiejowensis* remains, also accompanied by relatively a high diversified fossil assemblage. Vertebrates are represented by a second temnospondyl, *Cyclotosaurus intermedius Sulej & Majer, 2005*, a phytosaur (*Palaeorhinus*; see *Dzik, 2001*), the typical terrestrial tetrapod aetosaur *Stagonolepis olenkae Sulej, 2010*, pterosaurs (*Dzik & Sulej, 2007*), sphenodonts and other small tetrapods (*Dzik & Sulej, 2007*), as well as fishes (dipnoans described recently by (*Skrzycki, 2015*), and various actinopterygian and chondrichthyan species). Invertebrates, such as unionid bivalves (*Dzik et al., 2000*; *Skawina & Dzik, 2011*; *Skawina, 2013*), cycloids (*Dzik, 2008*), spinicaudatan crustaceans (*Olempska, 2004*), fresh-water ostracods (*Olempska, 2004*; *Olempska, 2011*), and some gastropods are also very common. The upper horizon is restricted to lenses cemented with

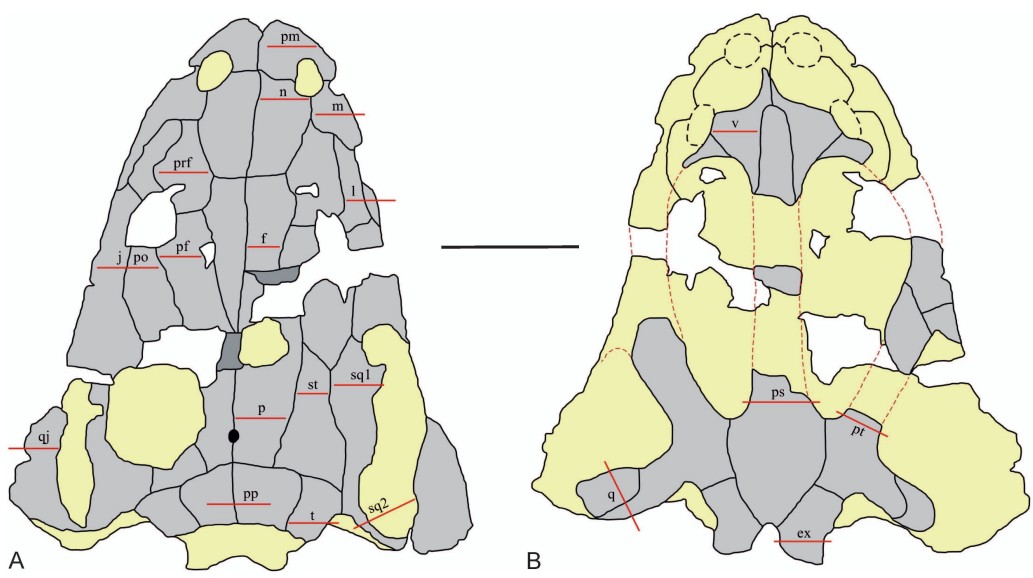

**Figure 2** **The sectioning planes of the *Metoposaurus krasiejowensis* skull (UOPB 01029) from the Late Triassic of Poland.** (A) The skull roof; (B) The palatal side of the skull. The sectioning planes are marked by red lines. Grey color indicates preserved parts of the skull; the destroyed or sediment-covered regions are indicated by the yellow color. Scale bar equals 10 cm. Abbreviations: pm, premaxilla; m, maxilla; n, nasal; l, lacrimal; prf, prefrontal; j, jugal; po, postorbital; pf, postfrontal; f, frontal; p, parietal; st, supratemporal; sq1, squamosal 1; sq2, squamosal 2; pp, postparietal; t, tabular; qj, quadratojugal; v, vomer; ps, parasphenoid; pt, pterygoid; q, quadrate bone; ex, exoccipital.

calcium carbonate, interpreted as a meander deposit (e.g., *Gruszka & Zieliński, 2008*). It is dominated by strictly terrestrial animals including *Stagonolepis* and the primitive dinosauromorph *Silesaurus opolensis* (*Dzik, 2003*). Aquatic vertebrates such as amphibians and phytosaurs are less common compared to their abundance in the lower horizon. Apart from that, one fragmentary specimen of the rauisuchian *Polonosuchus silesiacus* (*Brusatte et al., 2010*) was excavated between the upper and lower horizons (*Sulej, 2005*).

According to complex stratigraphic studies of the Upper Silesian Keuper, the bone-bearing beds were deposited in the early Norian times (*Racki & Szulc, 2014*; *Szulc, Racki & Jewupła, 2015a*; *Szulc et al., 2015b*). However, biochronological data uphold the Late Carnian age (e.g., *Dzik & Sulej, 2007*; *Lucas, Spielmann & Hunt, 2007*; *Lucas, 2015*).

## Methods

The skull was sectioned in 20 planes (Fig. 2), and the thin sections were prepared according to standard petrographic procedures (*Chinsamy & Raath, 1992*) in the Laboratory of the Adam Mickiewicz University in Poznań (Poland) and in the laboratory of Steinmann Institute (University of Bonn, Germany). The thin sections were ground and polished to a thickness of about 60–80 μm using wet SiC grinding powders (SiC 600, 800). Subsequently, the thin sections were studied under a LEICA DMLP light microscope in plane and cross polarized light.

The histological nomenclature follows, with an exception for annuli, *Francillon-Vieillot et al. (1990)* and *Witzmann (2009)*. According to *Francillon-Vieillot et al. (1990)*, the annual growth cycle consists of a thick, fast growing zone, a thin, slow growing annulus, and a
Line of Arrested Growth (LAG). In this study, the term zone is used as in its traditional meaning, for the highly vascularized layer with lower organization of collagen fibers. The term annulus, however (not following *Francillon-Vieillot et al., 1990*), refers to the less-vascularized bone with higher organization of collagen fibers, but usually similar in thickness as a zone. In the studied material, no clear LAGs can be observed. Instead, adjacent to the annuli, numerous lines are present. To avoid nomenclatural problems, all lines representing the cessation of growth are referred to in this study as resting lines, without determination of whether they occur annually or not (*Konietzko-Meier & Sander, 2013*; also see discussion).

In the thin sections, the average thickness of the entire bone and of each layer was estimated, expressed as an arithmetical average from three measurements of the thickness of the entire bone/layer. The thickness of the external layer was measured three times on the distance between the border line with middle region and the bottom of valleys, and three times as the distance between the border line (remodeling front) with middle region and the top of ridges. The mathematical average was calculated from these measurements. The minimum and maximum thicknesses represent the lowest and largest measurement, respectively, for each described layer. For the estimation of the ratio between the three components (E—external cortex, M—middle region, I—internal cortex; E:M:I), the average thickness of the external cortex was taken as one and then proportionately the value for middle region and internal cortex were calculated. Note that the internal cortex of dermal bone is oriented to the visceral surface of the body. Thus, in the parasphenoid, pterygoid and vomer, the external cortex is then oriented ventrally.

A detailed description of each bone is presented in the Supplemental Information 1.

## RESULTS

### Microanatomy of dermal bones

Most dermal bones of the skull are flat plates. Only the premaxillae and maxillae possess a more complicated shape (Fig. 3). The premaxilla is built up of three branches: the dental shelf, the alary process (*Schoch, 1999*), and the vomeral process, which connects the dental shelf to the vomer (Fig. 3A). The maxilla is built up from two branches: the dorsal one with an ornamented external cortex, and the ventral branch with the dental shelf (Fig. 3B).

The dermal bones show clear diploë (Fig. 4). The external cortex of the skull-roof bones created variably ornamentations built from a combination of grooves or tubercles and ridges (Table 1, Fig. 1), respectively visible in the cross-section as valleys and ridges (Fig. 4A). The thickness of the flat bones varies from under 1.5 to over 10 millimeters (Table 1), with different proportions between the particular layers. No constant relation can be observed between the thickness of the external cortex and the thickness of the entire bone. However, the external cortex of the tabular and postparietal, the two most massive bones, is clearly thicker than in other bones (Table 1). The relatively thin squamosal 2, with an average thickness of only about three mm, developed an external cortex which takes up almost half of the bone thickness (Table 1). The largest part of the bone almost always consists of the middle region (which is about two times thicker as the external cortex),

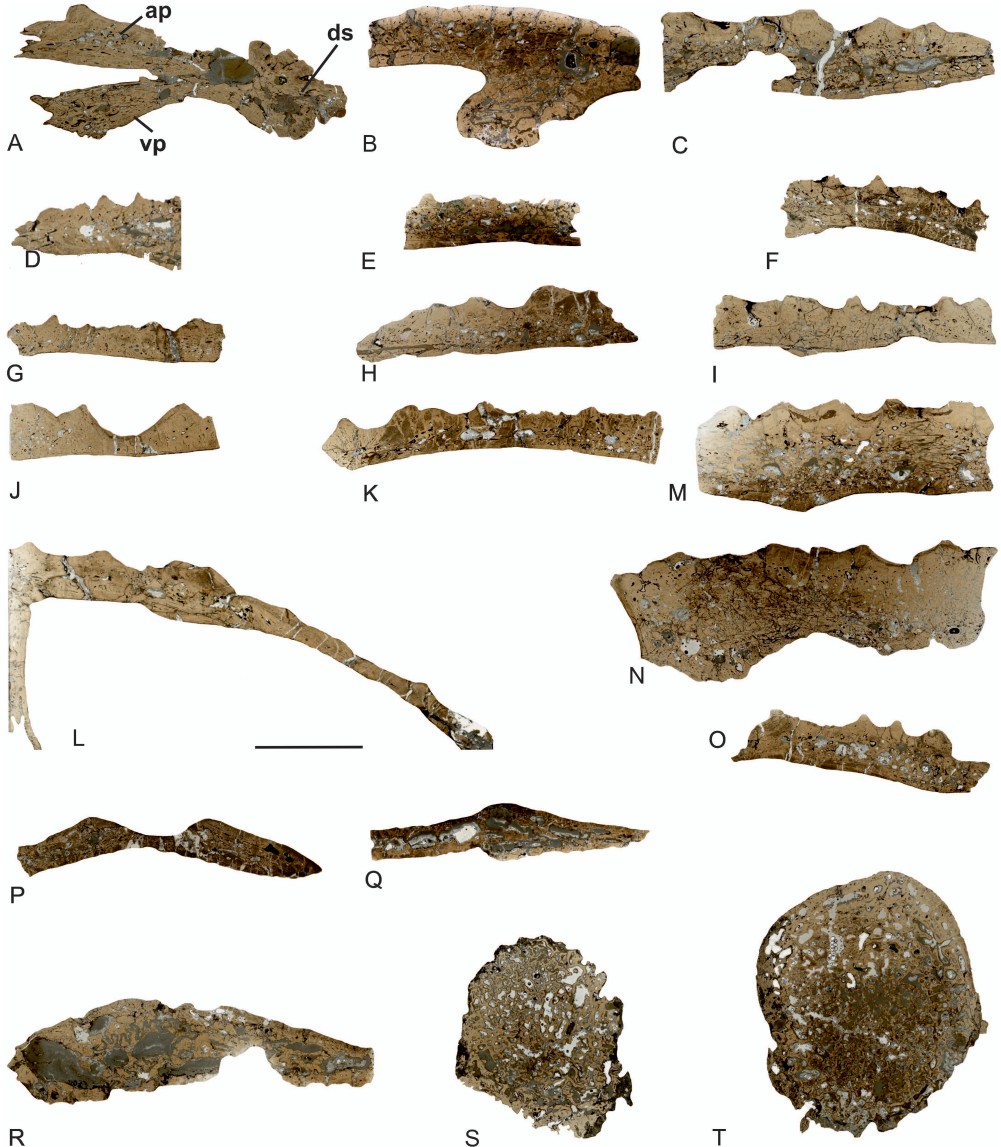

**Figure 3  General microanatomy of the skull bones of *Metoposaurus krasiejowensis* (UOPB 01029) from the Late Triassic of Poland.** (A) premaxilla; (B) maxilla; (C) nasal; (D) lacrimal; (E) prefrontal; (F) jugal/ postorbital; (G) postfrontal; (H) frontal; (I) parietal; (J) supratemporal; (K) squamosal 1; (L) squamosal 2; (M) postparietal; (N) tabular; (O) quadratojugal; (P) parasphenoid; (Q) vomer; (R) ptery-goid; (S) quadrate bone; (T) exoccipital. Scale bar equals 10 mm. Abbreviations: ap, alary process; ds, dental shelf; vp, vomeral process.

with the exception of squamosal 2, where the middle region is the thinnest (Table 1). The internal cortex is the thinnest of the three layers and composes usually 40%–90% of the thickness of the external cortex, with the exception of the pterygoid (Table 1).

## General histology of dermal bones
### *External cortex*
In all sections, the external cortex consists of parallel-fibered bone, whereas in the valleys lamellar bone often occurs (Figs. 4B and 4C). The elongated osteocyte lacunae with

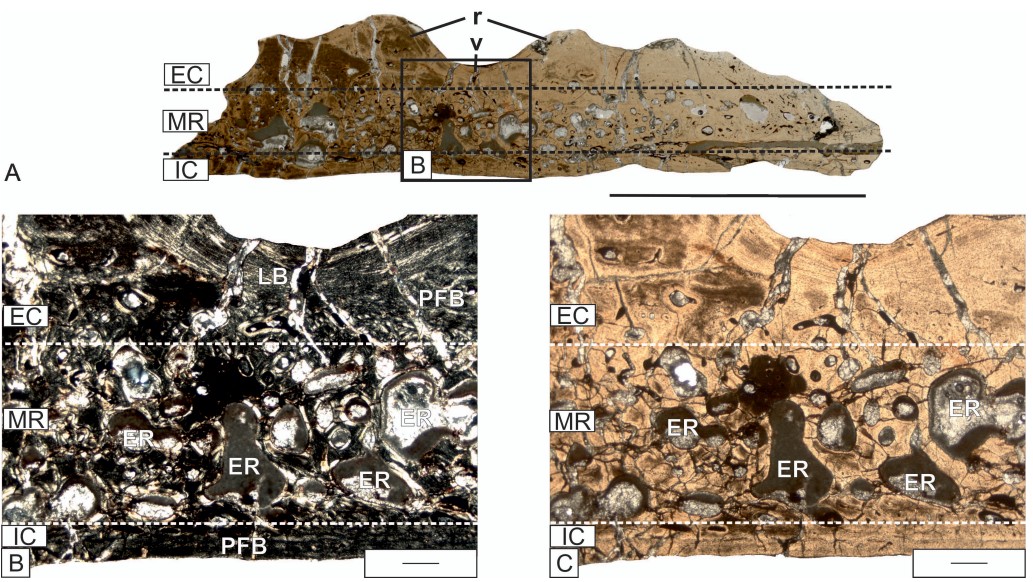

**Figure 4** **Detailed microanatomy of the skull bones of *Metoposaurus krasiejowensis* (UOPB 01029) from the Late Triassic of Poland, based on the frontal.** (A) A valley and two ridges; (B) Enlargement of (A); the external and internal cortex, and trabecular middle region with numerous and large erosion cavities are visible; image in cross-polarized light; (C) The same as (B), but in plane-polarized light. Dashed lines mark the approximately border between the external cortex/middle region/internal cortex. Scale bars equal 10 mm for (A), and 500 μm for (B–C). Abbreviations: EC, external cortex; ER, erosion cavities; IC, internal cortex; LB, lamellar bone; MR, middle region; PFB, parallel-fibered bone; r, rigde; v, valley.

branched canaliculi in the bone matrix are numerous (Fig. 5A). Vascular canals are mostly longitudinally oriented (Figs. 5A and 5B). The degree of vascularization varies from relatively low in the premaxilla and frontal, moderate in the maxilla and prefrontal, to highly vascularized in the jugal, postorbital, parietal, squamosal, quadratojugal, vomer and parasphenoid. In the nasal, postparietal, postfrontal, tabular and supratemporal, numerous vascular canals within the ridges are visible, which are arranged in rows parallel to the bony surface, whereas the valleys are avascular. The external cortex is dominated by simple vascular canals and primary osteons. In some bones (nasal, lacrimal, prefrontal, tabular, squamosal, vomer and parasphenoid), many secondary osteons and a few erosion cavities are visible in the transition region to the middle layer (for details see the Supplemental Information 1).

Typical for the external cortex are distinct collagen fibers (Figs. 5C and 5D). In the premaxilla, maxilla, nasal, lacrimal, jugal, postorbital, postparietal, and quadratojugal, well-mineralized Sharpey's fibers can be observed, which are relatively short but numerous, and sometimes packed densely in bundles. In the prefrontal, frontal, postfrontal, parietal, supratemporal, squamosal, tabular and vomer, Sharpey's fibers are rare and occur mostly in the deeper parts of the sculptural ridges (Fig. 5C). In the parasphenoid and pterygoid, Sharpey's fibers cannot be observed. In some bones (jugal, postorbital, postfronat, postparietal, tabular), thick fibers create Interwoven Structural Fibers (ISF) (Fig. 5D).

**Table 1  Microanatomy of the sampled bones of *Metoposaurus krasiejowensis* skull (UOPB 01029).**

| Bone | Ornamentation | Min-max thickness (μm) | Average thickness (μm)[a] | E:M:I[b] | Thickness of the external cortex (μm) |
|---|---|---|---|---|---|
| premaxilla–allary process (pm) | not preserved | 3,000–6,000 | 4,081.2 | 1:1.1:0.9 | 1,331.0 |
| maxilla (m)–dorsal process | not preserved | ~3,500–4,500 | ~3,060.0 | 1:1.1:0.9 | ~1,020.0 |
| nasal (n) | relatively high ridges (about 1,000 μm) | 4,500–7,000 | 4,758.0 | 1:1.2:0.4 | 1,830.0 |
| lacrimal (l) | medium high (500 μm), steep ridges and wide grooves | 4,500–6,500 | 5,940.0 | 1:1.6:0.7 | 1,800.0 |
| prefrontal (prf) | medium high (500 μm) and steep ridges | 3,900–5,000 | 4,256.0 | 1:1.7:0.8 | 1,150.0 |
| jugal (j) | high ridges (about 1,000 μm) and wide grooves | 5,000–8,000 | 6,940.0 | 1:1.5:0.7 | 1,800.0 |
| postorbital (po) | high ridges (about 1,000 μm) and wide grooves | 5,000–8,000 | 6,940.0 | 1:1.5:0.7 | 1,800.0 |
| postfrontal (pf) | low ridges (about 300 μm) and shallow grooves | 3,500–5,000 | 3,960.0 | 1:1.7:0.6 | 1,200.0 |
| frontal (f) | high ridges (about 1,000 μm) and wide grooves | 4,000–6,000 | 5,549.0 | 1:1.5:0.6 | 1,790.0 |
| parietal (p) | high ridges (about 1,000 μm) and narrow pits | 3,100–4,500 | 3,840.0 | 1:1.6:0.6 | 1,200.0 |
| supratemporal (st) | high ridges (about 1,500 μm) and wide grooves | 2,000–6,500 | 4,800.0 | 1:1.6:0.6 | 1,500.0 |
| squamosal 1 (sq1) | very high ridges (up to 2,000 μm) and wide grooves | 3,000–5,000 | 3,915.0 | 1:1.2:0.7 | 1,350.0 |
| squamosal 2 (sq2) | high ridges (about 1,000 μm) and wide grooves | 1,500–5,000 | 2,250.0 | 1:0.3:0.5 | 1,250.0 |
| postparietal (pp) | steep, high ridges (about 1,500 μm) and polygonal pits | 7,000–10,000 | 8,670.0 | 1:1.9:0.5 | 2,550.0 |
| tabular (t) | high ridges (about 1,000 μm) and wide pits | 7,000–11,000 | 10,000.0 | 1:2.2:0.8 | 2,500.0 |
| quadratojugal (qj) | high ridges (about 1,000 μm) and wide grooves | 4,000–6,000 | 5,610.0 | 1:1.7:0.6 | 1,700.0 |
| vomer (v) | no clear sculpture | 2,000–5,000 | 2,925.0 | 1:2.8:0.7 | 650.0 |
| parasphenoid (ps) | no clear sculpture | 2,000–4,700 | 4,050.0 | 1:2:1.5 | 900.0 |
| pterygoid (pt) | no clear sculpture | 4,500–7,000 | 5,460.0 | 1:5.4:2 | 650.0 |
| quadrate bone (q)* | – | diameter 20,000 μm | | | |
| exoccipital (ex)* | – | diameter 20,000 μm | | | |

**Notes.**

[a] The average thickness of entire bone was estimated in thin sections, expressed as an arithmetical average from three measurements of the thickness of a bone taken on the bottom of valleys and the top of ridges.

[b] For the estimation of ratio between external cortex (E), medial region (M) and internal cortex (I): E:M:I, the thickness of external cortex was taken as one and then proportionally the value for medial region and internal cortex were calculated.

*Non-dermal bone.

In the postparietal, a structure resembling metaplastic bone is constructed from longitudinally and transversely oriented structural fibers is visible (Fig. 5D).

Growth marks are expressed in two ways. In the ridges of the lacrimal, frontal, jugal, postfrontal, tabular, quadratojugal and squamosal 2, they are manifested as a sequence of

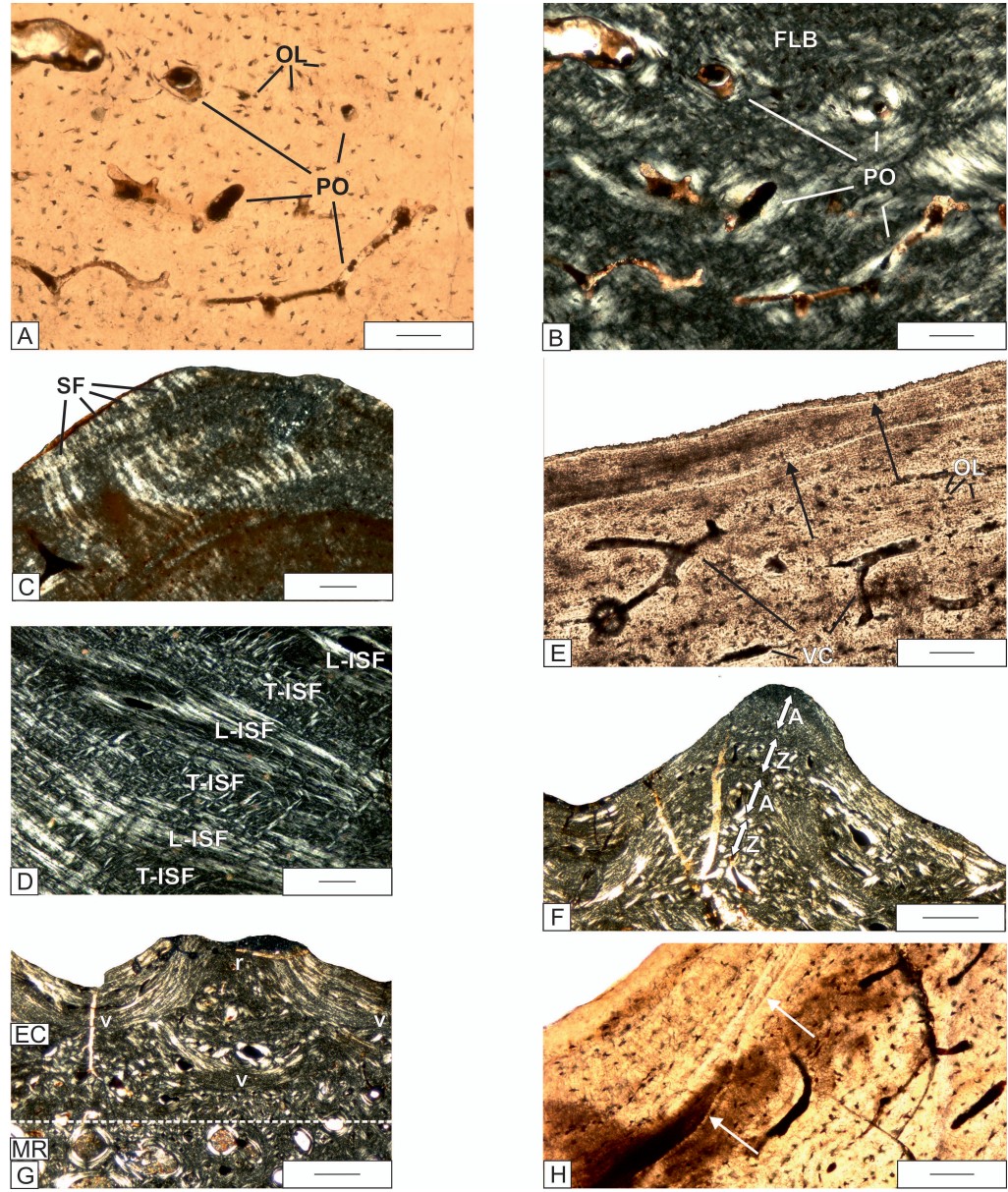

**Figure 5  Histology of the external cortex of the skull bones of *Metoposaurus krasiejowensis* (UOPB 01029) from the Late Triassic of Poland.** (A) Magnification of external cortex of the frontal; (B) Same as (A), but in cross-polarized light; (C) External cortex of the tabular with distinct Sharpey's fibers in the area of the sculptural ridges; (D) A succession of longitudinally and transversely cut ISF parts; (E) The resting lines (black arrows) in the cortex of squamosal 2; (F) Zones and annuli present in the external cortex of the quadratojugal; (G) Alternation of valleys and ridges in the postfrontal, note that remains of lamellar bone in the deep part of cortex are present, representing the bottom of a valley from an older generation; (H) The cementing lines (white arrows) visible in the superficial part of external cortex the supratemporal. Dashed lines mark the approximate border between the external cortex/middle region/internal cortex. Images (A), (E) and (H) in plane-polarized light, others in cross-polarized light. Scale bars equal for (A–E) and (H) 100 µm and for (F)–(G) 500 µm. Abbreviations: A, annulus; FLB, fibro-lamellar bone; L-ISF, longitudinally cut Interwoven Structural Fibers; OL, osteocyte lacunae; PO, primary osteons; r, ridge; SF, Sharpey's fibers; T-ISF, transversely cut Interwoven Structural Fibers; v, valley; Z, zone.

thin resting lines (Fig. 5E). In the quadratojugal, two thick annuli built up of lamellar bone alternate with two highly vascularized zones (Fig. 5F).

In the postfronal, squamosal 1, supratemporal, tabular, jugal and quadratojugal, the alternations of valleys and ridges are preserved. The remains of older valleys are filled with the highly vascularized tissue, which then constructed the ridges of the next generation (Fig. 5G). In the supratemporal, the cementing line is visible (Fig. 5H).

### Middle region

The external cortex changes gradually into the cancellous middle region. The simple vascular canals and primary osteons, of various shapes, are mostly located next to the border between the middle and external regions. A significant part of the middle region is strongly remodeled. The few large erosion cavities (up to 2,000 $\mu$m in diameter) are present in the premaxilla, nasal, lacrimal, jugal, prefrontal, postparietal, squamosal 1, quadratojugal and all studied bones from the palatal side of the skull. The most highly remodeled middle region occurs in the vomer, where the trabeculae are extremely reduced and erosion cavities in some areas exceed 3,000 $\mu$m in length (Fig. 6A). In the lacrimal and prefrontal, erosion cavities reach up to the external cortex. The maxilla is dominated by numerous, medium-sized erosion cavities. In the postfrontal, parietal and supratemporal, erosion cavities are small (less than 500 $\mu$m in diameter). In the postfrontal they appear sporadically, whereas they are numerous in the parietal and supratemporal (Fig. 6B). The middle region in the tabular and squamosal 2 does not show the typical trabecular structure. Intensive remodeling is visible; however, the tissue is relatively compact, almost without erosion cavities (Fig. 6C).

### Internal cortex

The internal cortex consists of parallel-fibered to lamellar bone. The degree of vascularization varies from very low and almost avascular in the parietal, postfrontal, supratemporal and squamosal, low in the premaxilla, prefrontal, nasal, and postparietal, moderate in the maxilla, frontal, vomer and parasphenoid, to high in the lacrimal, jugal, postorbital and quadratojugal (Figs. 6D–6F). Osteocyte lacunae, showing slightly elongated shapes, are very common. Growth marks are visible in the form of resting lines and a sequence of zones and annuli. The number of lines varies from four in the postfrontal and parietal, three in the postparietal, supratemporal and jugal, to two in the nasal (Fig. 6D). In the parasphenoid, well developed zones and annuli can be observed (Fig. 6G). Zones are built of thick, well vascularized layers, whereas annuli are represented by thinner, avascular layers. The numerous Sharpey's fibers packed in bundles are visible in the tabular and vomer.

## Endochondral bones
### Quadrate

The partially preserved and well-vascularized cortex consists of parallel-fibered and lamellar bone (Figs. 7A and 7B). The simple vascular canals occur sporadically, and secondary osteons are more common (Figs. 7A and 7B). The Sharpey's fibers are very short and occur only in the subsurface parts of the cortex. The elongated osteocyte lacunae are
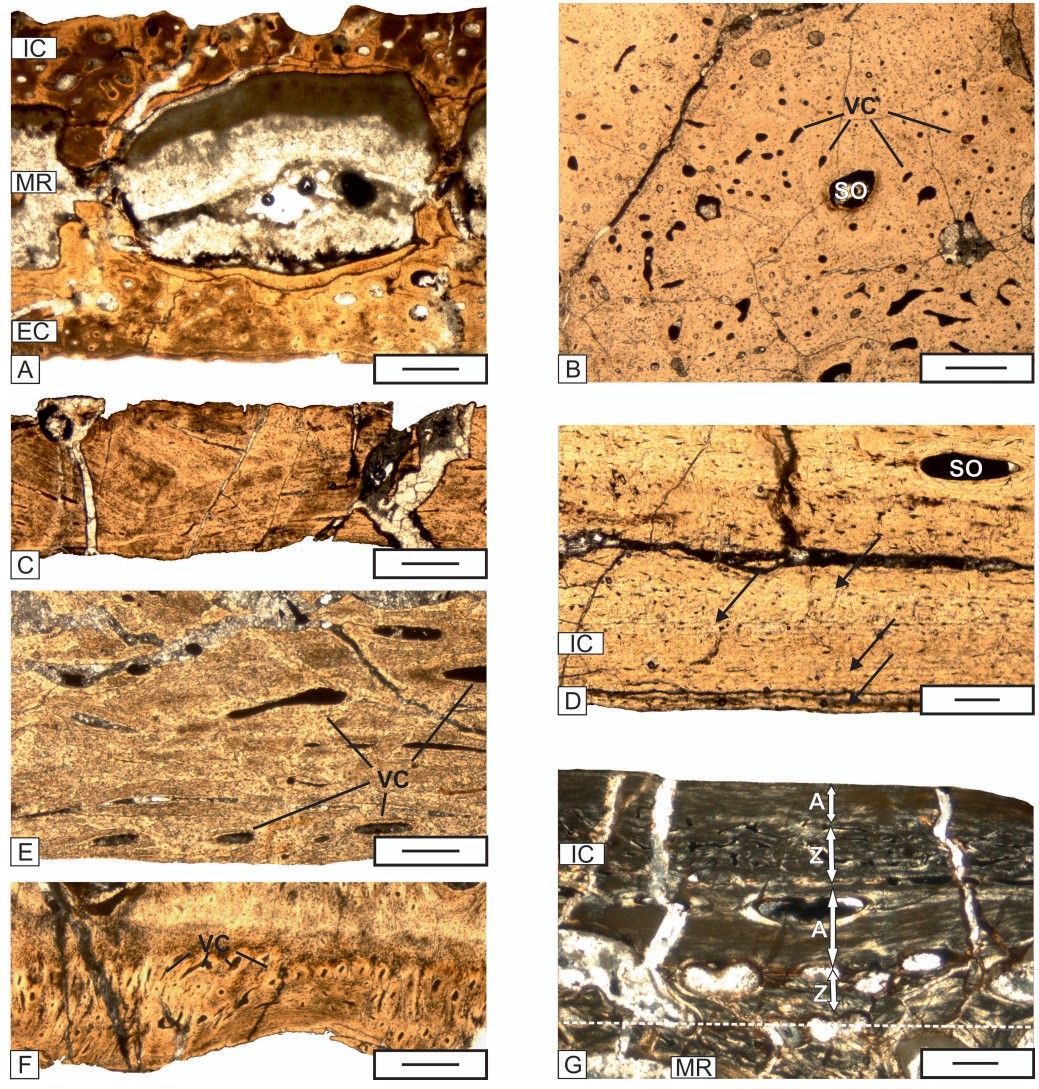

**Figure 6** **The details of the histology of the middle region and internal cortex of the skull (UOPB 01029) bones of *Metoposaurus krasiejowensis* from the Late Triassic of Poland.** (A) Large erosion cavities present in the middle region of the vomer; (B) Poorly remodeled, well vascularized middle region of the parietal; (C) Poorly vascularized fragment of the squamosal 2; (D) Almost avascular internal cortex with resting lines (black arrows) visible in the parietal; (E) Internal cortex of the premaxilla, note the relatively numerous vascular canals; (F) Internal cortex of the jugal with very numerous vascular, small vscular canals; (G) Alternation of thick annuli and zones visible in the internal cortex of the parasphenoid. Dashed lines mark the approximately border between the external cortex/middle region/internal cortex. Image (G) in cross-polarized light, others in plane-polarized light. Scale bars equal 100 $\mu$m for (D) and (G) and 500 $\mu$m for other photographs. Abbreviations: A, annulus; EC, external cortex; IC, internal cortex; MR, middle region; SO, secondary osteon; VC, vascular canals; Z, zone.

present mainly within the lamellar bone, which outlines the osteons. They do not possess canaliculi. Growth marks cannot be observed.

The central region consists of spongiosa and is characterized by large pore spaces and irregular trabeculae (Fig. 7C), which contain clumps of calcified cartilage (see also the Supplemental Information 1).

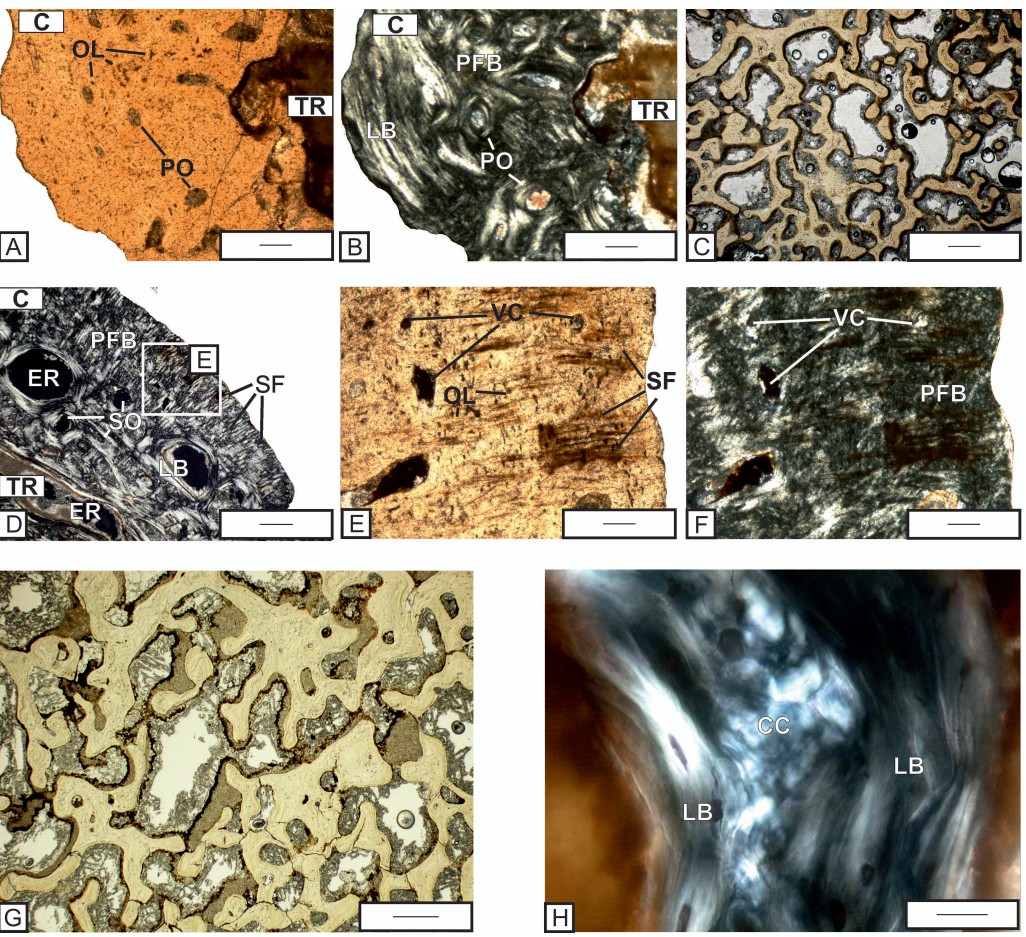

**Figure 7 Histological details of the quadrate (A–C) and exoccipital (D–H) of *Metoposaurus krasiejowensis* skull (UOPB 01029) from the Late Triassic of Poland.** (A) Fragment of cortex of the quadrate; (B) The same as (A), but in cross-polarized light; (C) Trabecular bone of the quadrate bone; (D) Fragment of cortex of the exoccipital with distinct Sharpey's fibers; (E) Close-up of (D), note that the Sharpey's fibers are also visible in plane-polarized light; (F) The same as (E) but in cross-polarized light. Both images (E) and (F) are rotated clockwise for better arrangement of the figures; (G) Trabeculae visible in the central part of the exoccipital; (H) Remains of calcified cartilage preserved in the trabeculae part of exoccipital. Images (A), (C), (E), and (G) in plane-polarized light, others in cross-polarized light. Scale bars equal 500 μm for (C), (D) and (G), and 100 μm for other photographs. Abbreviations: C, cortex; CC, calcified cartilage; ER, erosion cavities; LB, lamellar bone; OL, osteocyte lacunae; PO, primary osteons; PFL, parallel-fibered bone; SO, secondary osteons; SF, Sharpey's fibers; TR, trabecular region; VC, vascular canals.

## *Exoccipital*

The cortex consists of parallel-fibered bone and is relatively well-vascularized (Fig. 7D). The simple vascular canals are few in number (Figs. 7E and 7F) and located only in the outermost part of the cortex. The secondary osteons are more frequent (Fig. 7D). Well-mineralized, densely packed bundles of Sharpey's fibers are common and can be seen throughout the entire cortex (Figs. 7D and 7E). In the exoccipital, the Sharpey's fibers are most abundant and pronounced among all examined bones. Rounded osteocyte lacunae are numerous. Growth marks are absent.

The central region consists of an irregular network of bony trabeculae, with large pore spaces between them (Fig. 7G). In the medial parts of the bone tissue, where trabeculae are poorly developed, accumulations of calcified cartilage are quite common (Fig. 7H).

## DISCUSSION

### The histological variability

*Witzmann (2009)* investigated fragments of dermal bones from 20 taxa and concluded that for every taxon, the bone microanatomy and histology were consistent. Intraspecific variability of the histology of dermal bones was only observed in *Mastodonsaurus giganteus* and *Plagiosternum granulosum*, concerning the degree of vascularization and remodeling of the bone (*Witzmann, 2009*).

Nevertheless, in the *Metoposaurus krasiejowensis* skull, the variability is high and can be seen at both the microanatomical and histological levels. The bones pose variable thicknesses, different proportions between the layers, variations in the vascularization systems, tissue organizations, the presence and organization of Sharpey's fibers, degree of remodeling, and growth patterns (see also the Supplemental Information 1 for detailed descriptions). The combination of these characters shows that nearly very sectioning-plane in the skull represents a unique framework. The transition between the "histological types" is fluid. The jugal and postorbital, sectioned in the suture region, represent the same microanatomical and histological framework (Figs. 2 j and po, 3F; Table 1), whereas the squamosal sectioned in the frontal part of the bone (Fig. 2, sq1) and next to the otic notch (Fig. 2, sq2) show different architectures on both microanatomical and histological levels (Figs. 3K and 3L). This suggests that the histological framework is not specifically bone-limited, but seems to be related to the specific area of the skull. That is, it depends on the growth of the entire skull and each bone separately, the local biomechanical loading on a particular part of the skull, or a combination of both. *Fortuny et al. (2012)*, based on a Finite Elements Analysis, showed that the hypothetical biomechanical stress along the skull is different for each skull-morphotype and depends directly on the shape of the skull. Consequently, each taxon with a different skull morphotype responding to different loadings in a given region might have a unique histological architecture of homologous bones.

### Extent of remodelling

Among the sections from the *Metoposaurus* skull, four main degrees of remodeling could be observed in the middle region. Samples with the lowest degree of remodeling are from the postfrontal, parietal and supratemporal. A few large erosion cavities occur in the premaxilla, nasal, lacrimal, jugal, prefrontal, postparietal, squamosal, quadratojugal, and all studied bones from the palatal side of the skull. The maxilla is dominated by numerous, but moderately sized, erosion cavities. In the middle region of the tabular and squamosal 2, the bone deposition exceeds the bone resorption and it does not represent the typical spongious structure.

The increase in remodeling extent is a known developmental character. *Witzmann (2009)* published the detailed histology of dermal bones from a young adult and adult

*Mastodonsaurus,* and observed an increase in remodeling (expressed as an increase of the erosion cavities sizes) in the older specimen. In the *Metoposaurus*, different histological stages can be observed among different bones in one skull. The less remodeled (postfrontal, parietal and supratemporal) and highly remodeled (premaxilla, nasal, lacrimal, jugal, prefrontal, postparietal, squamosal, quadratojugal) bones seem to represent two stages of the same process, resulting in the increase in porosity of the middle region. This may indicate the sequence of the skull ossification during ontogeny, with the latest ossification of bones occurring on the central part of the skull roof. However, less remodeled samples originate from the grooves-ridges regions, whereas the other sections come from the reticulate areas. This confirms the hypothesis presented first by *Bystrow (1935)*, that the polygonal reticulate structures are the center of ossification and that ridges-grooves areas show the direction and extent of growth from these ossification centers. In this case, different degrees of remodeling, which resemble the ontogenetic change, are the result of longitudinal growth of the bone.

## Origin and dynamic of the sculpture pattern

Although dermal sculpture was early recognized as a characteristic of basal tetrapods (e.g., *Von Meyer, 1858*; *Fraas, 1889*; *Fritsch, 1889*; *Zittel, 1911*), the morphogenesis of the sculptures is still an open question (summarized by *Witzmann 2009*; *Witzmann et al. 2010*; *De Buffrénil et al. 2016*). Among extant tetrapods, growth of dermal bony tubercles and ridges has been studied in osteoderms of squamates and in dermal skull bones and osteoderms of crocodiles. In squamates, the presence of pits and ridges on the external surface of osteoderms follows from both local resorption and growth of bone (*Zylberberg & Castanet, 1985*; *Levrat-Calviac & Zylberberg, 1986*), whereas in crocodile dermal bones, *De Buffrenil (1982)* and *Cerda et al. (2015)* stated that sculpture is mainly the result of local resorption. In contrast, *Vickaryous & Hall (2008)* found no evidence for morphogenesis of bone sculpture by resorption in *Alligator mississippiensis*, and presumed that sculptural ridges develop by preferential bone growth. Concerning basal tetrapods, *Bystrow (1935)*; *Bystrow (1947)* showed that the development of bone sculpture in the temnospondyls *Benthosuchus*, *Platyoposaurus* and *Dvinosaurus* took place solely by growth of the bony ridges and tubercles, and resorptive processes were not involved. The thin sections of the dermal bones of skull and pectoral girdle in the basal tetrapods investigated by *Witzmann (2009)* corroborate Bystrow's findings and show that the dermal sculpture did not develop by local resorption of the bone surface, comparable to the pattern in basal tetrapod osteoderms (*Witzmann & Soler-Gijón, 2008*). According to the most recent study (*De Buffrénil et al., 2016*), the involvement of several complex remodeling processes, with the local succession of resorption and reconstruction cycles, is frequent and occurs in all major gnathostome clades, whereas the temnospondyl sections share an important common feature: the lack of superficial remodeling (resorption and reconstruction cycles). However, in the section of *Plagiosternum* described by *Witzmann (2009)*, the eroded external surface is illustrated. The supratemporal of *Metoposaurus krasiejowensis* (Fig. 5H) confirms the observation of *Witzmann (2009)* and shows that the remodeling process might be involved in the sculpture creation of Temnospondyli.

Moreover, the study of *De Buffrénil et al. (2016)* showed that, beside the resorption, other dynamics processes also modify the sculpture during bone growth. *De Buffrénil et al. (2016)* observed six main patterns of such modification. The simplest one is repetition of the width or position of pits and ridges from one growth stage to the following one. The ridges during the bone deposition can drift symmetrically in two opposite directions, or the ridges around a given pit may migrate in the same direction. Also, a change in size of the ridges is possible, resulting in the gradual narrowing of pit diameter (convergent ridge drift), or opposite process may occur when the reduction of ridge width is observed. In the most drastic case the pits can be entirely filled and disappear to be replaced *in situ* by ridges.

In the skull bones of *Metoposaurus krasiejowensis*, newly deposited bone repeats the pattern of sculptures in the younger stages (Figs. 5D–5F). However, in the postfrontal, squamosal, supratemporal, tabular, jugal, and quadratojugal, the alternation of valleys and ridges is preserved (Fig. 5G). In this case the newly deposited ridges are created on the place of valleys, but without resorption. The distance between newly created tops of the ridges is not distinctively different from that in the previous generation.

Overall, the evidence indicates that the metric pattern of the sculpture is relatively stable, but the position of the ridges and grooves is dynamic during growth as a specific adaptation to different biomechanical loading on the new, larger bone.

## Skeletochronological information

Long bones generally provide the best information for bone skeletochronological studies (*Castanet et al., 1993*; *Chinsamy-Turan, 2005*; *Erickson, 2005*). This also applies in temnospondyls (*Damiani, 2000*; *Steyer et al., 2004*; *Ray, Mukherjee & Bandyopadhyay, 2009*; *Mukherjee, Ray & Sengupta, 2010*; *Sanchez et al., 2010a*; *Sanchez et al., 2010b*; *Konietzko-Meier & Klein, 2013*; *Konietzko-Meier & Sander, 2013*; *Konietzko-Meier & Schmitt, 2013*; *Sanchez & Schoch, 2013*; *Konietzko-Meier, Danto & Gądek, 2014*). In the long bones, the three main types of growth marks are known: fast growing zones, more slowly deposited annuli, and Lines of Arrested Growth (LAG-s) that indicate the cessation of the growth (*Francillon-Vieillot et al., 1990*). Most often the full annual growth cycle consists of a thick, fast growing zone, a thin, slow growing annulus, and one or more LAG-s. Moreover, in fast-growing amniotes, the several growth lines (LAG-s) present next to the surface of bone, known as the External Fundamental System (EFS) could be visible. An EFS indicates a slowing of growth, suggesting that the maximum size has been reached (*Sander, 2000*; *Chinsamy-Turan, 2005*; *Erickson, 2005*; *Turvey, Green & Holdaway, 2005*; *Sander et al., 2011*). The dermal bones, e.g., osteoderms, have been used as well for skeletochronological analysis (*De Buffrénil & Buffetaut, 1981*; *Hutton, 1986*; *Hua & De Buffrénil, 1996*; *Tucker, 1997*; *Scheyer & Sander, 2004*; *Hill & Lucas, 2006*; *Hayashi & Carpenter, 2007*; *Scheyer, 2007*; *Scheyer & Sánchez-Villagra, 2007*; *Hayashi, Carpenter & Suzuki, 2009*; *Klein, Scheyer & Tütken, 2009*). However, the results of these studies suggest that a careful use of osteoderms in skeletochronology of fossil specimens is required because of different growth patterns between the skeleton and osteoderms (*Hayashi, Carpenter & Suzuki, 2009*; *Klein, Scheyer & Tütken, 2009*).
Even less is known about the preservation of growth marks in the temnospondyl dermal bones. Numerous growth marks in the external and internal cortices of the dermal bone have been observed in several temnospondyl taxa (*Scheyer, 2007*; *Witzmann, 2009*). However, without testing the whole growth series, it is not possible to estimate the amount of remodeled tissue and thus, no direct conclusion about the individual age of sectioned bones can be provided.

Histology of the long bones in *Metoposaurus krasiejowensis* is well known (*Konietzko-Meier & Klein, 2013*; *Konietzko-Meier & Sander, 2013*), and an evaluation and correlation of the growth patterns preserved in long bones and dermal bones is possible. Indirect estimation of the individual age of the studied skull is possible based on the morphological characters and size-comparison with the femora. Cranial sutures were not visible on the skull surface (*Gruntmejer, 2012*). The disappearance of all traces of sutures on the skull surface during ontogeny is a phenomenon often encountered in adult individuals (*Moazen et al., 2008*). In the completely preserved skeletons of *Dutuitosaurus ouazzoui* (*Dutuit, 1976*), a skull of similar length (about 400 mm) to the one here described, corresponds with an approximately 142 mm long femur (*Dutuit, 1976*: pl XXXI; D Konietzko-Meier, pers. obs., 2011). *Steyer et al. (2004)* calculated the individual age of the adult *Dutuitosaurus* femur, comparable in length, as approximately eight or nine years. Comparing skeletochronological data of *Metoposaurus* with that of *Dutuitosaurus* revealed that the femora of overlapping sizes show a similar age in both taxa, and major developmental plasticity can be excluded (*Konietzko-Meier & Klein, 2013*). The individual age of the Krasiejów skull, based on comparison with *Dutuitosaurus,* can be thus estimated at about eight to ten years.

In the *Metoposaurus* skull, two types of growth alternation can be observed: numerous resting lines in the external cortex in the lacrimal, frontal, jugal, postfrontal, postparietal, tabular, and quadratojugal, and alternation of thick zones and annuli in the external cortex of the quadratojugal (Fig. 5F) and in the internal cortex of parasphenoid (Fig. 6G). In *Metoposaurus* long bones, such aggregations of resting lines are present not only in the outer part of the cortex but also deeper. It suggests that accumulation of external resting lines does not mean the cessation of growth at all (EFS), but only the oscillation in growth rate during one season (*Konietzko-Meier & Klein, 2013*; *Konietzko-Meier & Sander, 2013*). The complex with accumulation of resting lines is interpreted as a single annulus deposited during one, dry season (*Konietzko-Meier & Sander, 2013*; *Konietzko-Meier & Klein, 2013*) and together with the more vascularized zone represents a full annual growth cycle. However, in the skull, resting lines occur only once in the outermost part of external cortex following the high vascularized layer. Without a growth series it is not possible to state if the pattern known from long bones applies also for dermal bones, and if older cycles have been already remodeled or if dermal bones show the independent growth outline. More informative are structures of the external cortex of the quadratojugal (Fig. 5F) and internal cortex in the parasphenoid (Fig. 6G), which resemble the growth sequence seen in *Metoposaurus* long bones (*Konietzko-Meier & Klein, 2013*; *Konietzko-Meier & Sander, 2013*). The two thick avascular layers represent the annuli, and combined with the two high vascularized zones indicate two growth seasons. Assuming that the age of the skull is about

eight or nine years, with the preservation of two growth cycles, the number of remodeled growth marks could reach up to six to seven. This indicates a relatively fast remodeling rate of the dermal bones of the skull compared to the long bones and confirms that the dermal bones are not a good source of skeletochronological information.

## Ossification processes of the skull

The quadrate and exoccipital show the periosteal ossification modus throughout the cartilage precursor. They consist of a trabecular middle region, surrounded by a thin layer of well-vascularized cortex. The preservation of cartilage (Fig. 7H), even during adulthood, indicates that ossification follows a pattern known for stereospondyl intercentra, with relatively slow ossification of the trabecular part and late development of the periosteal cortex (*Konietzko-Meier, Bodzioch & Sander, 2013*; *Konietzko-Meier, Danto & Gądek, 2014*).

The non-endochondral bones (dermal bones) may be formed through intramembranous ossification (dermal) or metaplastic ossification. Intramembranous ossification normally occurs in the deeper layers of connective tissue of the dermis of the skin (*Francillon-Vieillot et al., 1990*). Metaplastic bone develops via direct transformation of pre-existing, dense connective tissue, but in the absence of a periost, osteoblasts and osteoid (*Vickaryous & Hall, 2008*). Most often, the metaplastic and intramembranous domains occur together, creating many intermediate states between the intramebraneous bone, metaplastic bone, and even periosteal bone (*Main et al., 2005*). The metaplastic component of the dermal bone represents interwoven structural fibers (*Scheyer & Sander, 2004*; *Scheyer & Sánchez-Villagra, 2007*). In the *Metoposaurus* skull, interwoven structural fibers are found as islets or larger areas in the external cortex in all bones from the skull roof. Moreover, in the postparietal investigated here, the ridges are composed completely of structural fibers. The common occurrence of ISF suggests that metaplastic ossification plays an important role during cranial development. In contrast, the fragments of *Metoposaurus* bone described by *Witzmann (2009)* have an external cortex that is solely composed of well-ordered parallel-fibered bone with no metaplastic tissue. The lack of IFS may indicate that those bones sectioned by *Gross (1934)* do not belong to skull.

## Sharpey's fibers

In the long bones of *Metoposaurus*, the long Sharpey's fibers (SF1) indicate the remains of tendons and the shorter, very dense and evenly distributed fibers (SF2) are probably remains of bundles of collagenous fibers connecting periosteum to bone (*Konietzko-Meier & Sander, 2013*). In skull bones, also, both types of fibers could be recognized. In the prefrontal, frontal, postfrontal, parietal, supratemporal, squamosal, tabular and vomer, Sharpey's fibers are rare and occur mostly in the deeper parts of the sculptural ridges (Fig. 5C). In the premaxilla, maxilla, nasal, lacrimal, jugal, postorbital, postparietal, and quadratojugal, well-mineralized Sharpey's fibers are relatively short, but numerous, and sometimes packed densely in bundles. These fibers in cranial bone might represent tight anchorage of the dermis to the external bone surface, particularly to the sculptural ridges and tubercles, which served as the main points of anchorage for the skin.

The numerous long Sharpey's fibers packed in thick bundles are visible in the tabular. In this bone, the fibers occur also in the internal cortex. In the exoccipital, Sharpey's fibers

are densely packed in bundles, and they are much thicker and longer (Fig. 7D) than in the other bones. The Sharpey's fibers occur here in similar amounts to those described in vertebrae (*Konietzko-Meier, Bodzioch & Sander, 2013*). Large concentrations of long, well mineralized Sharpey's fibers in the tabular and exoccipital seem to be the obvious remains of strong muscle attachments and ligaments that connect the skull to the vertebral column.

## SUMMARY

Among the bone of the *Metoposaurus krasiejowensis* skull, the variability is very high and can be seen at both microanatomical and histological levels. The histological types are not specifically bone-limited, but seem to be related to the specific area of the skull. The observed pattern of remodeling progression suggests that the polygonal reticulate structures are the centers of ossification and that ridges-grooves areas show the direction and extent of growth from these ossification centers. The estimation of the individual age of the skull based on the morphological characters and comparison with the femora suggests a relatively fast remodeling rate of the dermal bones and confirms that the dermal bones are not a good source of skeletochronological information. The dynamic processes present in the external cortex (resorption and the alternation of the position of valleys and ridges) change the position of the ridges and grooves in what is a specific adaptation to different biomechanical loading on the new, larger bone.

Three main types of ossification occur in the skull. The quadrate and exoccipital show a periosteal ossification modus throughout the cartilage precursor. The preservation of cartilage, even during adulthood, indicates that ossification follows a pattern known for stereospondyl intercentra, with relatively slow ossification of the trabecular part and late development of the periosteal cortex. The non-endochondral bones (dermal bones) may be formed through intramembranous ossification (dermal) or metaplastic ossification. The common occurrence of ISF suggests that the metaplastic ossification plays an important role during the skull development. Short and dense Sharpey's fibers (SF2) visible in the external cortex are probably remains of tight anchorage of the dermis to the external bone surface. The numerous Sharpey's fibers packed in bundles, visible in the tabular and exoccipital, are the remains of strong muscle attachments and ligaments that connect the skull to the vertebral column.

## ACKNOWLEDGEMENTS

The authors acknowledge Olaf Dülfer (University of Bonn, Steinman Institute) for preparing some of the thin-sections. We are grateful to Kayleigh Wiersma (University of Bonn, Steinmann Institute) for improving the English. Aurore Canoville and Martin Sander are acknowledged for fruitful discussion. We thank both reviewers (Torsten Scheyer and Ignacio Cerda) and the editor (Andrew Farke) for all comments which greatly improved our paper.

### Funding
The authors received no funding for this work.

### Competing Interests
The authors declare there are no competing interests.

### Author Contributions
- Kamil Gruntmejer, Dorota Konietzko-Meier and Adam Bodzioch conceived and designed the experiments, performed the experiments, analyzed the data, contributed reagents/materials/analysis tools, wrote the paper, prepared figures and/or tables, reviewed drafts of the paper.

### Data Availability
The bone thickness measurements used in our manuscript and supplementary text are contained in Table 1. The size-descriptions of histological details (i.e., cell size or vascular canal diameter) is in the Results (general histology) and Supplemental Information 1.

### Supplemental Information
Supplemental information for this article can be found online at http://dx.doi.org/10.7717/peerj.2685#supplemental-information.

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
