# Peer review of "Cranial bone histology of Metoposaurus krasiejowensis (Amphibia, Temnospondyli) from the Late Triassic of Poland"

_PeerJ, doi:10.7717/peerj.2685_

## Round 0.1 · original submission · Major Revisions

- Reviewer 1 notes an issue with the interpretation of ISF (interwoven structural fibers), and suggests some reconsideration of the described morphology. This will require some adjustment of the interpretations, and must be addressed in revision.

- I strongly advise, but do not mandate, that the descriptions in the supplemental information be moved into the main body of the text. These are thorough and important data, and I fear they will be overlooked or ignored if relegated to supplemental information.

·

Basic reporting

Generally, I think the presented study is interesting and important to further our understanding of intraspecific and intra-individual variability of bone histological structures.

The English of the present version should be greatly improved - I made already quite some suggestions in the text, but I am probably missing much, not being a native speaker

Experimental design

Standard procedures are used to assess the bone histology. The images provided are generally of a high quality and suitable for this kind of study.

Validity of the findings

Some of the conclusions drawn by the authors are not sound I my view. I have marked the specific parts where I see the need for additional work or further scrutiny. This includes also the values and numbers provided in the main text, the table and the supplementary file.

Additional comments

I have made all comments directly on the PDF and the accompanying supplementary file (the two files are merged into one in the attached PDF).

·

Basic reporting

Dear Editor in Chief of PeerJ

I have reviewed the manuscript entitled “Cranial bone histology of Metoposaurus krasiejowensis (Amphibia, Temnospondyli) from the Late Triassic of Poland” by Kamil Gruntmejer and colleagues. They present a very interesting study about the bone microstructure of the skull bones of Metoposaurus. Since the destructive nature of histological methodology, this kind of studies on fossil vertebrates is really scarce and this contribution is really welcomed. The manuscript is well redacted; the descriptions are clear and concise. The figures of the main manuscript and supplementary files have been well prepared and exhibits good quality. The raw data is included in the supplementary files and allow to expand the general description of the main manuscript. Regarding the grammar, I’m no a native English speaker, but I can read and interpret the manuscript without major problems.
Although the issue is really interesting and deserves to be published in a high ranked journal as PeerJ, I found some problems that must be assessed before final acceptation of the paper.
*Introduction: It is clear that the objective of the paper is present the histological data on the skull of Metoposaurus, but it is not clear the main relevance of this. I think that the study is relevant for different issues, but the authors must explicit them. Since the authors actually discuss several paleobiological issues in the “Discussion” section, these subjects must be first present in the introduction (e.g. skeletochronology).

*Results: I found a mistake with regard to the histological description, which is actually important because the authors use this incorrect interpretation in the discussion. In the line 208 they write: “In the postparietal, a sequence of lamellar bone layers and layers consisting of ISF are visible (Fig. 5E).”. This is actually not seen in the figure. The term “ISF” refers to “Interwoven Structural Fibres’” and is used to describe coarse bundles of mineralized fibres, which are arranged in different direction forming a, precisely, interwoven pattern. In each bundle, the fibres are coarse and they orientate in parallel. When cut in parallel to the SF orientation, each bundle shows a general birefringence under polarized light (identical to the paralell fibred bone and similar to the lamellar bone). Conversely, when cut transversally, each bundle exhibits a general monorefringence under cross polarized light, and it is possible discern the limit between each single fibre (structural fibre in this case). In this case, each structural fibre is surrounded a very thin layer of birefringent tissue. This is actually the exact appearance of the “layer of ISF” that the authors describe in the text. Structural fibres are indeed present, but they do not form an interwoven fabric in the mentioned layer. From the figure 5.E it is very clear that ALL the tissue is formed by SF. These SF appears to be organized as “plywood”, in which the SF of the monorrefringent layer are oriented perpendicularly to the plane of section and, conversely, those located at the birrefringent layer (erroneously interpreted here as lamellar bone) are oriented parallel to the section plane. I strongly recommend to the authors to review the literature of osteoderm histology of vertebrates. They can found several examples of this kind of tissue in different vertebrates (really nice examples came from tryonichidae turtles [Scheyer et al. 2007.Org Divers Evol 7:
136–144) and sauropod dinosaurs [Cerda and Powell 2010, Acta Palaeontologica Polonica 55 (3): 389-398]). The incorrect interpretation has two main consequences in this study. First, the authors include them as growth marks, which is actually incorrect. Second, and more important, the presence of a plywood structure in the skull bone of Metoposaurus has very important implications with regard to the skull histogenesis (IS are formed by metaplastic ossification).
*Discussion: My main concern about the discussion is regarded with the fact that the authors have very interesting new data, but they does not provide a deep discussion on several aspects. I think that the papers will be really improved if the authors actually try to respond different questions. For example: Does metaplastic ossification contributes to the skull formation? (from the data, the response is YES), If actually contributes, the contribution is equal in all the bones?. Again, I encourage to the authors to expand the revision of those publications about dermal bone histology, not only in amphibia, but also other vertebrate groups. They will surely found very interesting ideas to include in this work, which provides really nice raw data.
These are my main concerns. There are other comments in the attached file.
On my opinion, the manuscript could be accepted only after a moderate revision.
I hope that the author found my review as a constructive contribution to its work.

Yours sincerelyIgnacio Cerda

Consejo Nacional de Investigaciones Científicas y Técnicas (CONICET)
Instituto de Investigación en Paleobiología y Geología
Universidad Nacional de Río Negro
Museo Carlos Ameghino, Belgrano 1700, Paraje Pichi Ruca (predio Marabunta) Cipolletti (8300)
Río Negro
Argentina

Experimental design

See "Basic reporting"

Validity of the findings

See "Basic reporting"

Additional comments

See "Basic reporting"

---

## Round 0.2 · Minor Revisions

Thank you for your close attention to the comments from the reviewers and me on the last version of the manuscript. I sent it out for a second round of review, and just a few minor suggestions were indicated by one of the reviewers. Additionally, I gave the manuscript a fairly thorough read-through, and noted a number of areas where style/grammar/syntax should be revised. I did not have time to do the same for the lengthy supplementary text, although would advise that this be done by someone (one of the authors or a colleague) prior to final acceptance, to ensure that everything is readable.

·

Basic reporting

The English has much improved and is acceptable now.

Experimental design

As in the first submission, standard procedures are used to assess the bone histology and the images provided are of a high quality and suitable for this kind of study.

Validity of the findings

My concerns given some of the interpretations have all been addressed by the authors. I do not have new comments on the revised version of the contribution.

·

Basic reporting

Dear Dr. Andrew Farke
I have reviewed the modified manuscript of Kamil Gruntmejer and colleagues. The authors have followed all my recommendations and I think that the manuscript has been highly improved. I suggest only few minor changes (introduced in the attached file) before final acceptation.


Best regards

Ignacio Cerda

Experimental design

See basic report

Validity of the findings

See basic report

Additional comments

See basic report

---

## Round 0.3 · Minor Revisions

Thank you for your attention to the previous round of comments. The manuscript is nearly ready for acceptance; in preparation for that, I gave the main manuscript file a thorough review for style and grammar. There are just a handful of minor corrections to make along this line. Please see the PDF uploaded with this decision; I will also send a .docx version to you directly for convenience in revising.

Once I receive your final edits, I should be able to approve the submission in very short order.

---

## Round 0.4 · accepted · Accept

Thank you for your attention to the final round of comments!